# Multidisciplinary Management of Rett Syndrome: Twenty Years’ Experience

**DOI:** 10.3390/genes14081607

**Published:** 2023-08-11

**Authors:** Sandra Vilvarajan, Madeleine McDonald, Lyndal Douglas, Jessica Newham, Robyn Kirkland, Gloria Tzannes, Diane Tay, John Christodoulou, Susan Thompson, Carolyn Ellaway

**Affiliations:** 1Faculty of Medicine, Sydney University, Sydney, NSW 2006, Australia; 2Department of Clinical Genetics, Children’s Hospital at Westmead, Sydney, NSW 2145, Australia; 3Department of Physiotherapy, Children’s Hospital at Westmead, Sydney, NSW 2145, Australia; 4Department of Occupational Therapy, Children’s Hospital at Westmead, Sydney, NSW 2145, Australia; 5Department of Speech Pathology, Children’s Hospital at Westmead, Sydney, NSW 2145, Australia; 6Department of Dentistry, Children’s Hospital at Westmead, Sydney, NSW 2145, Australia; 7Discipline of Child and Adolescent Health, The University of Sydney, Sydney, NSW 2006, Australia; 8Brain and Mitochondrial Research Group, Murdoch Children’s Research Institute, Melbourne, VIC 3052, Australia; 9Department of Pediatrics, University of Melbourne, Melbourne, VIC 3052, Australia; 10Genetic Metabolic Disorders Service, Children’s Hospital at Westmead, Sydney, NSW 2145, Australia

**Keywords:** Rett syndrome, *MECP2*, diagnosis, multidisciplinary management, clinical trials

## Abstract

Over the last 20 years, the understanding and natural history of Rett syndrome has advanced, but to date no cure has emerged, with multidisciplinary management being symptomatic and supportive. This study provides a comprehensive review of the clinical features, comorbidities and multidisciplinary management of a well-characterized cohort of females with classical Rett syndrome. We aim to improve awareness and understanding of Rett syndrome amongst pediatricians, pediatric subspecialists and allied health professionals to enable early diagnosis and a streamlined enrolment approach for future clinical trials. Rett syndrome, a complex X-linked condition, affecting mainly females, is due to pathogenic variants of the *MECP2* gene in most affected individuals. The Rett syndrome Multidisciplinary Management clinic at The Children’s Hospital at Westmead, Sydney, Australia, was established in 2000. This retrospective analysis of individuals who attended the clinic from 2000 to 2020 was performed to identify the incidence and predicted age of onset of Rett syndrome related comorbidities, disease progression and to review management principles. Data collected included age of Rett syndrome diagnosis, *MECP2* genotype, clinical features and medical comorbidities, such as sleep disturbance, seizures, breathing irregularities, scoliosis, mobility, hand stereotypies, hand function, constipation, feeding ability, use of gastrostomy, communication skills, QTc prolongation, anthropometry, and bruxism. Analysis of 103 girls who fulfilled the clinical diagnostic criteria for classical Rett syndrome with a pathogenic variant of the *MECP2* gene showed a median age of diagnosis of 3 years. The most frequent *MECP2* variant was c.502 C>T.

## 1. Introduction

Rett Syndrome (RTT; OMIM entry #312750) is one of the most common genetic causes of developmental and intellectual disabilityt in females, affecting up to 1:10,000 females under the age of 12 years [1]. RTT was first described [2] in 1966 as a neurodevelopmental disorder occurring in young females [2]. Classical RTT is characterized by a unique developmental regression period, particularly involving fine motor skills and communicative abilities, occurring after apparently normal early development with the onset of intense stereotypic midline hand movements [3]. Other clinical features include epilepsy, cognitive impairment, scoliosis, feeding difficulties, growth restriction, sleep disturbance, bruxism, autonomic and motor dysfunction [3]. Although the disorder is caused by a single gene, the features of classical RTT are quite distinctive and there is wide phenotypic variability. Some individuals have been recognized with many of the features of classical RTT, such as developmental regression, but do not have all of the features. These are classified as atypical RTT and include a preserved speech variant, early seizure variant and congenital onset variant [3].

Pathogenic, usually de novo variants of the X-linked, Methyl-CpG-binding protein 2 (*MECP2*) gene causes the majority of classical RTT and a smaller proportion of atypical RTT [4]. The diagnosis of classical RTT is based on a set of clinical diagnostic criteria [3]. Atypical RTT is diagnosed when an individual does not fulfill all of the essential criteria for classical RTT and represents the most and least severe forms of RTT [3]. Other genes such as *CDKL5* and *FOXG1* have been associated with atypical RTT or a RTT-like phenotype [5,6] and may manifest with preserved function and specific clinical features [3]. Some atypical variants have a specific *MECP2* variant, such as the preserved speech variant, which has a milder phenotype, with retained mobility and communication [6]. Studies of genotype–phenotype correlations have identified some *MECP2* variants resulting in less severe phenotypes in both classical and atypical RTT, such as p.Arg133Cys, Arg 294X, p.Arg306Cys and 3′ truncations. In contrast other variants, such as p.Arg 106Trp, p.Arg 168X, p.Arg 255X, p.Arg270X and deletions, have been associated with more severe phenotypes [7]. 

Despite its phenotypic variability, more than 95% of classical and approximately 75% of atypical RTT cases have a pathogenic variant in *MECP2*. *MECP2* encodes the methyl CpG binding protein 2 (MeCP2). MeCP2 is expressed throughout the body but is most abundant in mature neurons [8]. It binds to methylated cytosines in DNA through its methyl binding domain and, through the transcription repression domain, modulates transcription [8]. Genotype–phenotype correlation studies have identified some variants that are less severe than others and large deletions as the most severe [7]. It has been hypothesized that X chromosome inactivation may contribute to phenotypic variability [9]. *MECP2* variants have been identified rarely in males with a neurodevelopmental disorder termed “male RTT encephalopathy” [10]. 

The average age at RTT diagnosis has trended downwards with the improvement and increasing availability of diagnostic genetic testing. Children with RTT are living into adulthood, with approximately 60% living to early middle age [11]. Currently, there is only one drug treatment, Trofinetide, which is the analog of the terminal unit of insulin growth factor 1 (IGF1) recently approved by the FDA for RTT and only available in the USA, which has shown safety in large pediatric cohorts and clinical improvement in the neurobehavioral symptoms of RTT [12,13]. There are clinical trials in the pipeline, including gene therapy. In the meantime, management is symptomatic and supportive. Management of RTT requires a multidisciplinary approach due to its multisystem involvement. A multidisciplinary management clinic dedicated to RTT was established at The Children’s Hospital at Westmead, Sydney, Australia, in 2000. The clinic has a diverse team of medical practitioners, including a Pediatrician, Clinical Geneticist and Dentist, along with Allied Health professionals such as a Genetic Counsellor, Dietitian, Physiotherapist, Occupational Therapist, Speech Therapist and a Music Therapist. It is the only such clinic in the state of New South Wales, Australia, which has a population of over 8 million. 

This study analysed a cohort of females with classical RTT who attended the RTT multidisciplinary management clinic at the Children’s Hospital at Westmead, Sydney, Australia, over a 20 year period and aims to provide a comprehensive review of their clinical features, comorbidities and multidisciplinary management. 

## 2. Materials and Methods

### 2.1. Study Design

A single institution, retrospective chart review of 103 patients who attended the RTT MDT clinic from 2000 to 2019 inclusive was performed. Observations and assessments from MDT consultations were accessed from the electronic medical record as well as paper files from the clinic. Two researchers collected each patient’s data from each of their clinic visits. The data collected included age of RTT diagnosis, *MECP2* genotype, recorded deaths, any previous diagnoses, sleep disturbances, seizures, breathing irregularities, communication skills, QTc prolongation, growth data, bruxism, scoliosis, age of spinal surgery, leg length discrepancy, hip dysplasia, mobility, hand stereotypies, hand function, feeding skills, pubertal status, constipation and other GIT issues including gastrostomy placement. Data collection definitions are as defined below.

### 2.2. Ethics

This study was approved by the Sydney Childrens’ Hospital Network Human Research Ethics Committee, ETH13655. 

### 2.3. Participants

All individuals had an initial consultation with the Pediatrician/Clinical Geneticist and Genetic Counsellor to establish or confirm the diagnosis of classical RTT prior to being referred to the RTT MDT clinic. Patients were invited to attend the clinic every 1–2 years. 

One hundred and three females, aged less than 18 years with classical RTT and a pathogenic *MECP2* variant and who had attended the MDT clinic, were included in this study. The classification of the RTT phenotype was based on current clinical diagnostic criteria [3]. Patients classified as atypical RTT prior to 2010 due to the lack of deceleration of head growth [14] were reclassified as classical RTT according to the revised clinical diagnostic criteria and included in this dataset. Patients with atypical RTT, males, females over 18 years of age, with brain injury or other neurological disorder, who did not attend a MDT clinic and without a pathogenic *MECP2* variant were excluded from this study. 

### 2.4. Data Collection

Patient Medical records and Genetic files were accessed for data collection.

### 2.5. Genetics

The *MECP2* variant and protein change were recorded^14^. Less common *MECP2* variants were reported as ‘other variants’ or ‘other deletions’. 

### 2.6. Medical Comorbidities 

Breathing irregularities, bruxism, seizures, constipation, scoliosis, sleep disturbances, prolonged QTc, gastrostomy placement, hip dysplasia, leg length discrepancy and spinal surgery were recorded if they were reported at any visit, even if they resolved over time. Age of onset of seizures was based on the date of a diagnosis by a pediatric neurologist. The age of spinal surgery and gastrostomy placement were also recorded. 

### 2.7. Mobility, Hand Function and Communication Skills

Mobility was classified into three categories—immobile, i.e., wheelchair bound or only able to stand but unable to take steps, assistance, i.e., being able to take steps whilst being held by another person, and independent, i.e., able to take steps without any assistance. Hand stereotypies were recorded. Hand function was reported, from most able to least able as having pincer grip, palmar grasp, or no hand function. Communication was stratified into ability to say phrases (ability to string two or more words together), single words or being completely non-verbal. 

### 2.8. Growth and Feeding

Height/length and weight were recorded at each MDT clinical visit. Z scores and BMI Z scores were also calculated and recorded using age appropriate female standard growth charts (CDC). The International Dysphagia Diet Standardization Initiative (IDDS) framework classification [15] was used as a reference range for food and beverage consistency. The levels of assistance required for feeding was assessed using the Eating and Drinking Ability Classification System (EDACS) [16] for level of assistance. This is a validated tool for use in Cerebral Palsy [16].

### 2.9. Statistical Analysis

All statistical analyses was performed using SPSS version 27. Frequency tables were generated for genotype, medical comorbidities and feeding history. Descriptive statistics including mean, standard deviation (SD), median, interquartile range (IQR), range and skewing were performed for the age of diagnosis of RTT, age of onset of seizures and age of spinal surgery gastrostomy insertion and for Z scores. Z scores for the RTT were compared with standard values using paired *t*-test. Generalized estimating equations explored the impact of multiple observations per girl.

## 3. Results

### 3.1. Number of Clinic Visits

The number of visits to the Rett syndrome Multidisciplinary Managament clinic for each patient were recorded Figure 1. 

### 3.2. Diagnosis

Between January 2000 and December 2019, 103 girls diagnosed with classical RTT based on the clinical diagnostic criteria and a pathogenic *MECP2* variant were seen in the MDT clinic. The median age of diagnosis, either clinically or by molecular testing, was 3 years (IQR 2–4 years), with ages ranging from 1 to 16 years. An initial diagnosis of autism spectrum disorder (ASD) was reported in 10 (9.7%) girls. Table 1 shows the *MECP2* variants of the cohort. 

### 3.3. Medical Comorbidities

Table 2 shows the medical comorbidities within the RTT cohort. Bruxism was the most commonly reported co-morbidity, occurring in 77%. Seizures were reported in 74%, with a median age of onset of 4 years (IQR 3–6.75 years of age). Constipation was reported in 65% and all required specific treatment with aperients. Scoliosis was diagnosed in 61 girls (59.2%) with 12 (11.7%) requiring spinal surgery at a median age of 12.5 years (IQR 12–15 years). Prolonged corrected QT value was identified in 32%. Further investigation of those with a prolonged QT value with a 24 h Holter monitor study did not identify any cardiac arrythmias. 

Gastrostomy feeding tube placement was required for 24 (23.3%) girls at a mean age of 8 years (SD 4.2 years). Indications for gastrostomy feeding tube placement included deterioration in feeding abilities, poor weight gain, dysphagia and poor oral intake. One individual had significant aerophagia and abdominal distension that interfered with feeding and gastrostomy was used to vent the stomach. In addition, gastroesophageal reflux (GORD) was reported in 17 (16.5%) and coeliac disease was diagnosed in 2 (1.9%). Premature adrenarche was identified in 9 (8.7%) girls and precocious puberty in 2 (1.9%). There were 12 (11.7%) deaths recorded over the 20 year period.

### 3.4. Breathing Irregularities 

Breathing irregularities were observed and/or reported in 71 (68.9%) of the cohort. This most commonly included intermittent episodes of breath-holding and/or hyperventilation. Table 3. 

### 3.5. Mobility

The ability to walk independently was observed in 47 (45.6%) of the cohort. Deterioration in mobility with age was observed in 16 (15.5%) Figure 2. 

### 3.6. Hand Stereotypies and Function

Hand stereotypies, including midline clasping, clapping, mouthing, tapping, and wringing, were observed in all girls and 13 (12.6%) lost all previously attained hand function. All exhibited intentional eye gaze directed upper limb movement, such as hand to mouth, clapping, banging or swiping, when provided with a motivation such as a musical instrument, food or drink. This could be facilitated by holding one limb to prevent stereotypic hand movements. Figure 3.

### 3.7. Communication

Verbal communication was retained in 27 (26.2%) girls, but this was limited to single words or short phrases, often used without context. Twenty two (21.4%) girls had lost all previously attained verbal skills. All girls used non-verbal communication skills, such as the use of body language, eye gaze, gestures, and vocalizations Figure 4. 

### 3.8. Growth and Feeding

At first visit, height/length, weight and BMI were significantly skewed towards lower percentiles than the standard population (Table 4). Over time, there was a trend for Z scores for height and weight to decline with age. 

Feeding history was available for 100 patients with the majority (85.4%) fully orally fed but tolerating a range of food/liquid consistencies. Of 267 total observations, the following textures were consumed: all food textures (19.7%), soft, bite sized food (38.9%), minced moist food (22.7%) and pureed or liquids (25.4%). There was no clear trend in texture tolerance with age. The level of assistance needed with feeding was high with 64.4% recorded visits showing full assistance with feeding was required and in only 9.4% of visits were the girls independent in feeding. 

## 4. Discussion

Retrospective analysis of a cohort of females with classical RTT has highlighted multisystem comorbidities and the requirement for multidisciplinary management. As demonstrated in previous studies, we show that classical RTT is frequently associated with bruxism, seizures, constipation, scoliosis, prolonged QTc and sleep disturbances. In addition all patients exhibited the hallmark midline hand stereotypies, loss of hand function and spoken language and dyspraxic gait or immobility. 

Molecular analysis of the cohort revealed that the four most frequently occurring *MECP2* variants were p.Arg168*, p.Arg133Cys, p.Arg294* and p.Thr158Met, which are consistent with previously reported *MECP2* hotspot variants [17]. 

The median age of diagnosis was three years of age and reflects the increasing knowledge and awareness of RTT, as well as greater access to diagnostic genetic testing. 

Bruxism was the most widely observed symptom, affecting 76.7% of girls, which can result in attrition and loss of tooth structure, sensitivity and difficulties for carers in maintaining oral hygiene. Our clinical experience has found some reduction in bruxism following the eruption of secondary dentition and the risk of dental caries is not significantly higher in RTT.. The management of bruxism in RTT is mainly conservative. 

Previous studies [11] have documented the common association of seizures in RTT. All individuals in this cohort with seizures had their first seizure before 15 years of age with a median age of seizure onset of 4 years. 

An abnormal breathing pattern is commonly associated with RTT as confirmed by this study. The majority of our cohort had intermittent episodes of hyperventilation and/or breath-holding and most exhibited a combination of abnormal breathing patterns. Anecdotally, parents reported an increase in hyperventilation in times of excitement, anxiety, stress or as an expression of pain. Hyperventilation and breath-holding can lead to central cyanosis, but rarely requires medical intervention and resolves spontaneously. These incidents are thought to be related to the autonomic dysfunction associated with RTT [18]. It is important for clinicians, particularly in hospitalization settings, to refrain from unnecessary intervention. 

With regard to motor ability, 47 (45.6%) girls retained independent mobility and 16 (15.5%) had a deterioration in mobility. A previous multidisciplinary clinical review observed that only half the girls were independent mobilisers [19] compared with 45.6% of girls who retained independent mobility in our cohort. The gait of RTT individuals has been reported on observation as [20] toe walking, wide based gait, rocking and/or ataxic and all were observed in this study. Hypotonia may be present in the first few years of life and can evolve to hypertonia and rigidity with increasing age [19]. General muscle tone was assessed at each clinic visit. Changes in mobility may correlate with changes in muscle tone and the development of orthopedic complications such as scoliosis, which was reported in 59.2%. Some girls were reported to have a deterioration in mobility following the onset of seizures. Physiotherapy intervention is recommended to encourage greater independence with ambulation, reduce agitation and prevent the development or worsening of contractures and other secondary musculoskeletal complications.

Retention of some limited verbal communication was observed in 27 (26.2%) girls, which included the use of single words or short phrases, not all used in context. All girls exhibited non-verbal communication skills, such as the use of body language, gestures and eye gaze, particularly for choice making. 

Intense midline stereotypic hand movements such as wringing, tapping, mouthing and clasping are the hallmark of RTT. Hand function, particularly hand grasp in our cohort,, was observed to deteriorate with age. Some individuals, however, retained limited hand skills, such as finger feeding, holding a drink bottle, using a loaded fork and choice making. At times, intense hand stereotypies can be reduced to allow functional movement with the use of elbow splints or the holding of one hand. In our RTT MDT clinic, music therapy complemented the assessment of hand function and movement and allowed communication, emotional expression and reduced anxiety. Music therapy has been shown to improve cognitive and physical development [21].

Deceleration in growth and weight is well reported in RTT [22,23], as is seen in this cohort, with BMI considered appropriate as a longitudinal measure of nutritional status [24]. Poor nutritional status may, however, occur given feeding difficulties, and gastrostomy feeding tube placement may be indicated [25] for nutritional support, as occurred in 23% of our population 

Our study demonstrates similar results to previous studies [26] which have shown QT prolongation to be present in RTT. QT prolongation is a potentially life threatening condition and can lead to sudden death. No patient with QT prolongation in our cohort was found to have cardiac arrhythmias on subsequent Holter monitor studies. However, clinicians should be cautious in prescribing medications known to alter QT intervals in patients with RTT. 

Sleep disturbances were commonly noted amongst the girls, but there was inconsistency of reporting in the medical records. Disturbances that are common in RTT include night waking, nighttime laughter and screaming and have no significant correlation to age [27]. Melatonin was the most common medication prescribed in our cohort and is most helpful for sleep initiation [28]. Other medications commonly prescribed for management of sleep disturbance included clonidine, chloral hydrate, Phenergan and risperidone [28].

During the 20 year period, 12 individuals died. Identifying the cause of death was beyond the scope of this study, as deaths occurred outside the Children’s Hospital at Westmead and were inconsistently reported in the medical records. 

### Study Limitations

The RTT MDT clinic is based at a pediatric tertiary hospital where adults are not seen. One of the limitations of this study, therefore, was the age of our cohort being limited to 18 years, and lack of data once individuals are transitioned to adult services. Being a specialized clinic, the patient group may be biased towards those with more complex medical problems related to the diagnosis of RTT. However, the review was conducted in a specialized MDT clinic rather than with parent/carer questionnaires or surveys. 

The results of this study highlight the usefulness of our clinical model based on multidisciplinary management aimed at preserving and/or prolonging independence in mobility, feeding and communication. Many studies evaluating frequencies of comorbidities rely on small sample sizes followed over short time periods. Clinical reviews by multidisciplinary clinics [11,19] are highly valued as they provide longitudinal data within a large, well-defined cohort to further ascertain the frequency and nature of the comorbidities in RTT. This review provides a holistic overview of RTT management and allowed achievement of statistically significant results with a larger cohort. 

## 5. Conclusions

RTT is a rare, complex monogenic neurodevelopmental disorder with multisystem involvement. This study showed the prevalence of multiple comorbidities, most prominently of which are breathing abnormalities, mobility and speech impairment, diminished hand function, bruxism, seizures, constipation, scoliosis and sleep disturbances. The median age of diagnosis was 3 years old, showing the increasing awareness of clinical education regarding RTT and improved access to genetic diagnostics. There is evolution of symptoms and co-morbidities over time and with lifelong functional dependence. Individuals with RTT require lifelong personal care, assistance and suitable supportive equipment. The type of support will vary over the RTT lifespan for each individual and their families/carers. The accumulation of knowledge regarding the natural history of RTT in well-defined cohorts serves as a valuable resource for health care providers. This study adds further to the literature, providing a greater understanding of RTT in a cohort over two decades. We anticipate that this will be helpful to assist clinicians in the diagnosis, counselling and management of RTT. It is imperative that clinicians recognize RTT early to avoid delayed or misdiagnosis and enable appropriate early multidisciplinary care. Future clinical trials will be reliant on the rapid identification of well-defined patient cohorts and centres of clinical expertise. 

## Figures and Tables

**Figure 1 genes-14-01607-f001:**
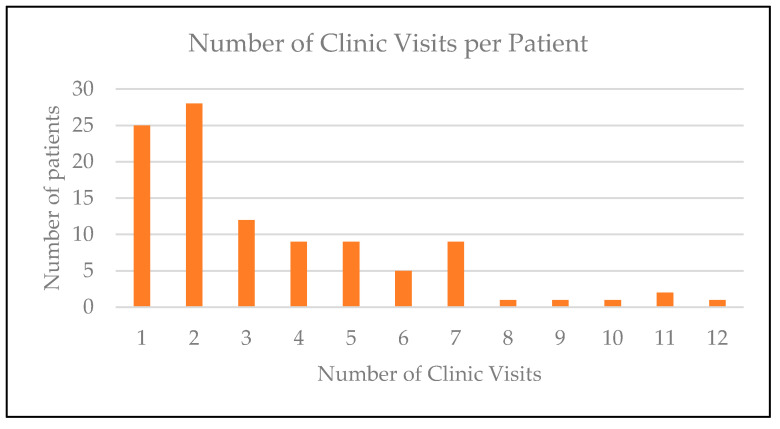
Histograph showing number of clinic visits per patient.

**Figure 2 genes-14-01607-f002:**
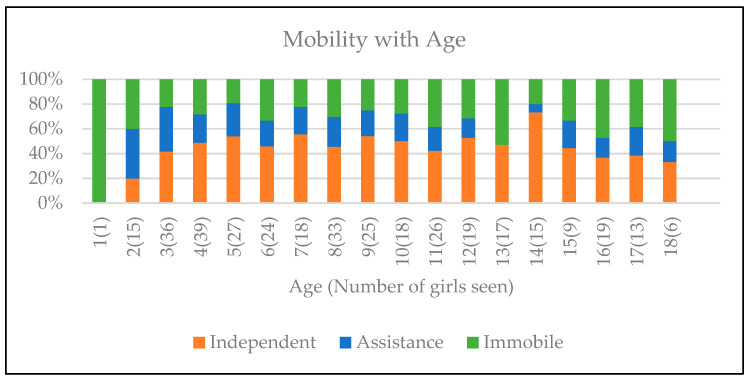
Changes in mobility correlated to age. Brackets indicate the number of girls at each age.

**Figure 3 genes-14-01607-f003:**
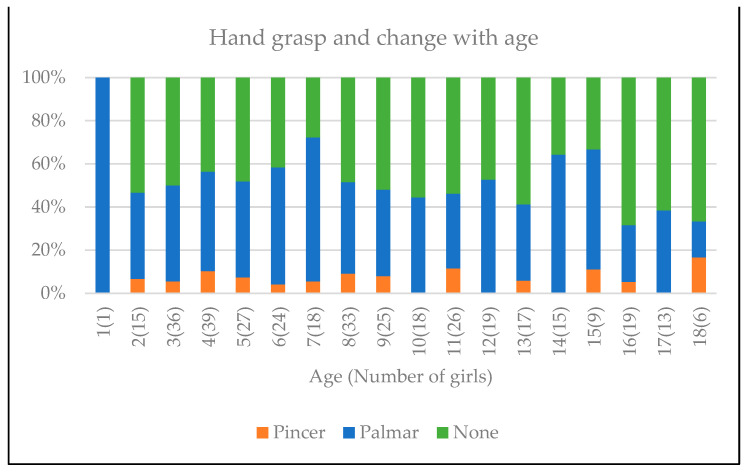
Changes in hand grasp correlated to age. Brackets indicate the number of girls at each age.

**Figure 4 genes-14-01607-f004:**
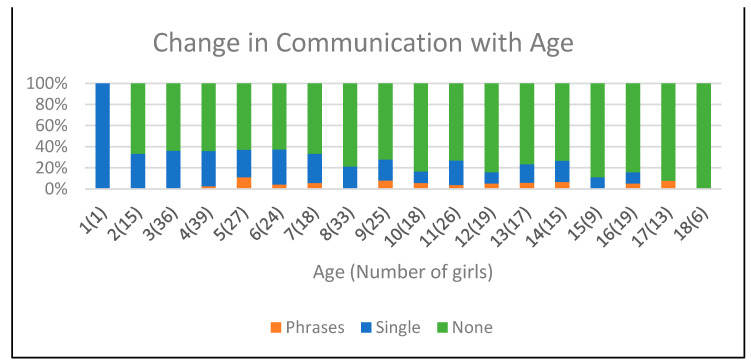
Changes in communication correlated to age. Brackets indicate the number of girls at each age.

**Table 1 genes-14-01607-t001:** *MECP2* variants reported in the cohort.

cDNA	Protein	*n* (%)
c.316 C > T	p.Arg106Trp	3 (2.9)
c.397 C > T	p.Arg133Cys	10 (9.7)
c.455 C > G	p.Pro152Arg	3 (2.9)
c.473 C > T	p.Thr158Met	7 (6.8)
c.502 C > T	p.Arg168 *^,1^	14 (13.6)
c.763 C > T	p.Arg255 *	5 (4.9)
c.808 C > T	p.Arg270 *	4 (3.9)
c.806delG	p.Gly269fs	3 (2.9)
c.880 C > T	p.Arg294 *	8 (7.8)
c.916 C > T	p.Arg306Cys	4 (3.9)
c.1164_1207del44	p.Pro389 *	2 (1.9)
c.1157_1197del41	p.Leu386fs	4 (3.9)
Other deletions		19 (18.4)
Other variants		17 (16.5)
Total		103 (100)

^1^ * = Stop Codon.

**Table 2 genes-14-01607-t002:** Medical co-morbidities.

Symptom	*n* (%)
Bruxism	89 (76.7)
Seizures	76 (73.8)
Constipation	67 (65.0)
Scoliosis	61 (59.2)
Sleep Disturbances	50 (48.5)
Prolonged QTc	33 (32.0)
Gastrostomy	24 (23.3)
Hip Dysplasia	22 (21.4)
Leg length Discrepancy	14 (13.6)
Spinal Surgery	12 (11.7)

**Table 3 genes-14-01607-t003:** Incidence of breathing patterns.

Breathing Pattern	*n* (%)
Breath Holding	53 (51.5)
Hyperventilation	53 (51.5)
Aerophagia	10 (9.7)
Forced Expulsion of Air	28 (27.2)

**Table 4 genes-14-01607-t004:** Weight, height and BMI expressed as Z scores ^1^ of standard growth charts at first visit (mean yrs. + STD).

	*n*	Mean	Standard Deviation (Min, Max)	Lower, Upper 95% CL for Mean	Pr > |t|
Height/Length	97	−1.04	1.39 (−4.79, 2)	−1.32, −0.76	<0.001
Weight	102	−1.17	1.97 (−6.55, 3.34)	−1.56, −0.78	>0.001
BMI	97	−0.54	1.72 (−5.39, 3.44)	−0.89, −0.19	0.0026

^1^ Z score of 0 is equivalent to 50th centile while a Z score of 2 plots at around the 98th centile and −2 at the 3rd centile.

## Data Availability

The data generated by this research project is subject to privacy and ethical restrictions.

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
