# Peer review of "Multidisciplinary Management of Rett Syndrome: Twenty Years’ Experience"

_genes, 2023, doi:10.3390/genes14081607_

Round 1

Reviewer 1 Report

The manuscript of Vilvarajan provides the retrospective analysis of individuals affected with Rett syndrome with a special highlight on the comorbidities, genetic variants and disease progression. In the introduction some aspects could have been defined (e.g. atypical RTT). Methods are well described and results are understandable. The figures in color would be better. There is a limitations section that highlights many crucial aspects.

The manuscript is an important contribution to the long natural history studies in genetic diseases.

Comments:

1. Atypical RTT should be defined in the first paragraph

2. Line 61: please give example of the genes involved in atypical RTT

3 Paragraph 2-are there any other genotype-phenotype correlations?

4 Line 78- Please describe this treatment in a few words

5.  In the methods please state clearly how many patients have been included

6.  I don’t find paragraph 3.1 and Figure 1 informative. They could be deleted.

7. Table 1-"other deletion" is it meant that the genetic variants in unknown? 

8. Figures would look better in colors

9. Some parts of the discussion would fit better to the results section e.g. “We identified 9 girls with premature adrenarche and 2 with precocious puberty”.

10. Discussion could be shortened and comparison to other studies illustrated in the table

11. Conclusions section could be also shortened and more to the point, not including perspectives but more focusing on the study done

Author Response

Responses to Reviewer 1

  1. Abstract:The abstract need to be arranged and some details and a clear conclusion at the end. Please put the aim at the first part.
    1. The abstract has been changed to include the Aim at the beginning and Conclusion at the end.

  1. Introduction: Please summarize the introduction and also the aim should be more clear and at the end of the introduction
    1. Aim added to the end of the introduction.

  1. The study methodologyis NOT well described.  The Materials and Method sections are incomplete. The inclusion criteria should be clearer
    1. Edited materials/methods section.

 We have added further information about the inclusion and exclusion criteria for this study.

  1. Due to the various comorbidities evaluated in this study the methods section was broken down to adequately describe the endpoints under ‘data collection’ rather than condensed under ‘study design’.

  1. Results and discussion– very good. Except Table 3 is doubled. Please correct the table numbers.
    1. Corrected

  1. The references need to be adjusted according to the journal style.
    1. Changed to MDPI as per journal style.

Reviewer 2 Report

The paper is technically sound. The data support the conclusions. However, it needs revision:

1.     Abstract: The abstract need to be arranged and some details and a clear conclusion at the end. Please put the aim at the first part.

2.     Introduction:

Please summarize the introduction and also the aim should be more clear and at the end of the introduction.

3.     The study methodology is NOT well described.  The Materials and Method sections are incomplete. The inclusion criteria should be clearer

4.     Results and discussion – very good. Except Table 3 is doubled. Please correct the table numbers.

5.     The references need to be adjusted according to the journal style.

The paper is technically sound. The data support the conclusions. However, it needs revision:

1.     Abstract: The abstract need to be arranged and some details and a clear conclusion at the end. Please put the aim at the first part.

2.     Introduction:

Please summarize the introduction and also the aim should be more clear and at the end of the introduction.

3.     The study methodology is NOT well described.  The Materials and Method sections are incomplete. The inclusion criteria should be clearer

4.     Results and discussion – very good. Except Table 3 is doubled. Please correct the table numbers.

5.     The references need to be adjusted according to the journal style.

Author Response

Responses to Reviewer 2

  1. Atypical RTT should be defined in the first paragraph
    • We have added a definition of atypical RTT.

  1. Line 61: please give example of the genes involved in atypical RTT
    • This has now been included in line 72 where is is most relevant.

  1. Paragraph 2-are there any other genotype-phenotype correlations?
    • We have included further information regarding genotype phenotype studies

  1. Line 78- Please describe this treatment in a few words
    • Added description of drug treatment

  1. In the methods please state clearly how many patients have been included
    • This has been added to the ‘study design’ paragraph and ‘participants’ paragraph.

  1. I don’t find paragraph 3.1 and Figure 1 informative. They could be deleted.
    • We haven’t removed Figure 1 at this stage as we feel it puts context to the number of total data points that were analysed and included

  1. Table 1-"other deletion" is it meant that the genetic variants in unknown? 
    • This refers to other MECP2 variants ie specifically deletions (not included in the table) and other variants.
    • Please note that all individuals in this cohort had a known MECP2 variant

  1. Figures would look better in colors
    • Colours added

  1. Some parts of the discussion would fit better to the results section e.g. “We identified 9 girls with premature adrenarche and 2 with precocious puberty”.
    • Moved to results section

  1. Discussion could be shortened and comparison to other studies illustrated in the table
    • We have made some adjustments as suggested but we feel that it is important to cover all aspects/comorbidities for his cohort in keeping with the aims of the study

Please note that the editors have requested we lengthen the paper rather than shorten it.

  1. Conclusions section could be also shortened and more to the point, not including perspectives but more focusing on the study done
    • More focus on the study has been added.

Please note that the editors have requested we lengthen the paper rather than shorten it.

Round 2

Reviewer 1 Report

Thank you for the revised manuscript. I am satisfied with the revised manuscript version.